# Kolmogorov–Smirnov GAN

## Abstract

We propose a novel deep generative model, the Kolmogorov-Smirnov Generative
Adversarial Network (KSGAN). Unlike existing approaches, KSGAN formulates
the learning process as a minimization of the Kolmogorov-Smirnov (KS) distance,
generalized to handle multivariate distributions. This distance is calculated using
the quantile function, which acts as the critic in the adversarial training process.
We formally demonstrate that minimizing the KS distance leads to the trained
approximate distribution aligning with the target distribution. We propose an
efficient implementation and evaluate its effectiveness through experiments. The
results show that KSGAN performs on par with existing adversarial methods,
exhibiting stability during training, resistance to mode dropping and collapse, and
tolerance to variations in hyperparameter settings. Additionally, we review the
literature on the Generalized KS test and discuss the connections between KSGAN
and existing adversarial generative models.

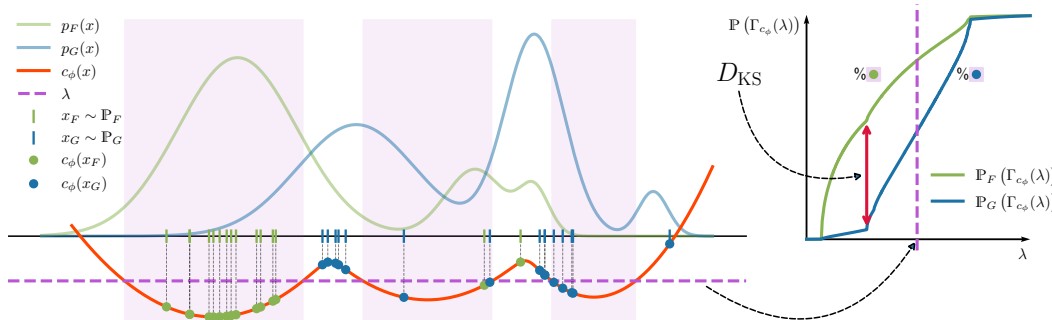

Figure 1: A schematic depiction of how the Generalized Kolmogorov-Smirnov (KS) distance between
target $\mathbb{P}_F$ and approximate $\mathbb{P}_G$ distributions with respect to critic $c_\phi$ is computed. The critic is
evaluated on samples $x_F$ (|) and $x_G$ (|) from the target and approximate distributions respectively.
The $\lambda$ threshold moves from $-\infty$ to $+\infty$ establishing a stack of level sets. At each level, the fraction
of datapoints (● and ●) below the threshold is calculated for each distribution independently. This
produces the $\mathbb{P}_F\left(\Gamma_{c_\phi}(\lambda)\right)$ and $\mathbb{P}_G\left(\Gamma_{c_\phi}(\lambda)\right)$ curves. The Generalized KS distance is the largest
absolute difference between the curves shown as ↕ in the right figure. Best viewed in color.

## 1 Introduction

Generative modeling is about fitting a model to a target distribution, usually the data. A fundamental
taxonomy of models assigns them into *prescribed* and *implicit* statistical models [9], with partial
overlap between the two classes. Prescribed models directly parameterize the distribution's probability

density function, while implicit models parameterize the generator that allows samples to be drawn from the distribution. The ultimate application of the model primarily dictates the choice between the two approaches. It does, however, have consequences regarding the available types of divergences that we can minimize when fitting the model. The divergences differ in the stability of optimization and computational efficiency, as well as statistical efficiency, which all affect the final performance of the model.

The natural approach for fitting a prescribed model is maximum likelihood estimation (MLE), equivalently formulated as minimization of Kullback–Leibler divergence. Likelihood evaluation for normalized models is straightforward. In non-normalized models, density evaluation is expensive; in this context, Hyvärinen [22] proposed the score matching objective, which can be interpreted as the Fisher divergence [30]. This approach is very effective for simulation-free training of ODE[7]/SDE[42, 19]-based models which are state-of-the-art in multiple domains today.

The principle driving the fitting of implicit statistical models is to push the model to generate samples that are indistinguishable from the target. An inflection point for this family of models came with the Generative Adversarial Network (GAN) [13], which took the principle literally and introduced an auxiliary classifier trained in an adversarial process to discriminate between the two distributions. The classification error given an optimal classifier relates to the Jensen–Shannon divergence between generator and the target. Initial work in this area involved applying heuristic tricks to deal with learning problems, namely vanishing gradients, unstable training, and mode dropping or collapse. Further advancements focused on using other distances based on the principle of adversarial learning of auxiliary models, which were supposed to have certain favorable properties with respect to the original GAN.

The Bayesian inference community has been reluctant to adopt adversarial methods [8], and the attempts to apply them in this context [40] indicate a credibility problem. A significant drawback of approximate methods is the excessive reduction of diversity in the distribution [17], the extremes of which lead to mode dropping [1]. In this work, we consider another distance for training implicit statistical models, i.e., the Kolmogorov-Smirnov (KS) distance, which, to the best of our knowledge, has not been used in this context before. The distinctive feature of the KS distance is that it directly measures the coverage discrepancy of each other's credibility regions by the distributions under analysis at all confidence levels. Thus, its minimization straightforwardly leads to the correct spread of the probability mass, avoiding mode dropping, overconfidence, and mode collapse when applied with a sufficient sampling budget.

We term the proposed model as *Kolmogorov-Smirnov Generative Adversarial Network* (KSGAN). We show how to generalize the standard KS distance to higher dimensions based on Polonik [38] in section 2, allowing our method to be used for multidimensional distributions. Next, in section 3, we show how to efficiently leverage the distance in an adversarial training process and show formally that the proposed algorithm leads to an alignment of the approximate and target distributions. We support the theoretical findings with empirical results presented in section 6.

## 2 Generalized Kolmogorov–Smirnov distance

We generalize the Kolmogorov–Smirnov (KS) distance (sometimes called simply Kolmogorov distance) between continuous probability distributions on one-dimensional spaces to multidimensional spaces and show that it is a metric. The test statistic of the KS test is a KS distance between empirical and target distributions (or two empirical in the case of the two-sample case). For this reason, our proposal is directly inspired by the generalization of the test introduced in Polonik [38].

Let us consider two probability measures $\mathbb{P}_F$ and $\mathbb{P}_G$ on a measurable space $(\mathcal{X}, \mathcal{A})$, where the sample space $\mathcal{X}$ is a vector space such as $\mathbb{R}^d$ and $\mathcal{A}$ is the corresponding event space; $F : \mathcal{X} \to [0, 1]$ and $G : \mathcal{X} \to [0, 1]$ are the cumulative distribution functions (CDFs) of $\mathbb{P}_F$ and $\mathbb{P}_G$ respectively.[1] We say that $\mathbb{P}_F = \mathbb{P}_G$ iff $\forall A \in \mathcal{A}, \ \mathbb{P}_F(A) = \mathbb{P}_G(A)$. When $\dim(\mathcal{X}) = 1$ then the KS distance is

$$D_{\text{KS}}(\mathbb{P}_F, \mathbb{P}_G) := \sup_{x \in \mathcal{X}} |F(x) - G(x)|. \tag{1}$$

In the multivariate case, the problem with using the KS distance as is is that on a $d$-dimensional space, there are $2^d - 1$ ways of defining a CDF. The distance has to be independent of the particular

---

[1]In what follows we will use $\mathbb{P}_F$ for the true data distribution and $\mathbb{P}_G$ for the learnt one

definition and thus should be the largest across all the possibilities [35]. This, however, becomes prohibitive for any $d > 2$. In other words, the challenge comes from a multidimensional vector space not being a partially ordered set. Everything that follows in this section consists of proposing a partial order, showing that, under certain conditions, a probability distribution can be uniquely determined on its basis and operationalizing it in an optimization problem.

We begin by bringing the classical result that

$$D_{\text{KS}}(\mathbb{P}_F, \mathbb{P}_G) = \sup_{\alpha \in [0,1]} |F(G^{-1}(\alpha)) - \alpha|, \tag{2}$$

where $G^{-1} : [0, 1] \to \mathcal{X}$ is the inverse CDF also called the quantile function. Einmahl and Mason [10] show that there exists a natural generalization of the quantile function to multivariate distribution, which we restate below.

**Definition 1** (Generalized Quantile Function). *Let $\text{v} : \mathcal{A} \to \mathbb{R}_+$ be a measure, and $\mathcal{C} \subset \mathcal{A}$ an arbitrary subset of the event space, then a function $C_{\mathbb{P},\mathcal{C}}(\alpha) : [0, 1] \to \mathcal{C}$ such that*

$$C_{\mathbb{P},\mathcal{C}}(\alpha) \in \arg\min_{C \in \mathcal{C}} \{\text{v}(C) : \mathbb{P}(C) \geqslant \alpha\} \tag{3}$$

*is called the generalized quantile function in $\mathcal{C}$ for $\mathbb{P}$ with respect to $\text{v}$.[2]*

The generalized quantile function evaluated at level $\alpha$ yields a *minimum-volume set* [36] whose probability is at least $\alpha$, and it is the smallest with respect to $\text{v}$ such set in $\mathcal{C}$, thus the name. For the remainder of this paper, we assume that $\text{v}$ is the Lebesgue measure.

It may seem that it is enough to plug $C_{\mathbb{P}_G,\mathcal{C}}(\alpha)$ in place of $G^{-1}(\alpha)$ and $\mathbb{P}_F$ in place of $F$ in eq. (2) to establish the Generalized KS distance but it turns out that such a distance does not satisfy the positivity condition $D_{\text{KS}}(\mathbb{P}_F, \mathbb{P}_G) > 0$ if $\mathbb{P}_F \neq \mathbb{P}_G$ as the example below shows.

**Example 1** (Polonik [38]). *Let $\mathbb{P}_F$ be the probability measure of a chi distribution with one degree of freedom $\sqrt{\chi_1^2}$ which has support on $\mathbb{R}_+$ and $\mathbb{P}_G$ the probability measure of a standard Gaussian distribution $\mathcal{N}(0, 1)$ which has support on the whole $\mathbb{R}$. Given $\mathcal{C} = \mathcal{A}$ we have*

$$\mathbb{P}_F(C_{\mathbb{P}_G,\mathcal{C}}(\alpha)) = \alpha \,\forall \alpha \in [0, 1], \tag{4}$$

*while clearly $\mathbb{P}_F \neq \mathbb{P}_G$. The statement in eq. (4) is easy to show by observing that $\forall x \in [0, \infty)$ the density of $\mathbb{P}_F$ is twice the density of $\mathbb{P}_G$ and $C_{\mathbb{P}_G,\mathcal{C}}(\alpha)$ are intervals centered at 0.*

Instead, a solution based on the quantile functions of both distributions is needed, which we present in definition 2.

**Definition 2** (Generalized Kolmogorov-Smirnov distance). *Let the Generalized Kolmogorov-Smirnov distance be formulated as follows:*

$$D_{\text{GKS}}(\mathbb{P}_F, \mathbb{P}_G) := \sup_{\substack{\alpha \in [0,1] \\ C \in \{C_{\mathbb{P}_G,\mathcal{C}}, C_{\mathbb{P}_F,\mathcal{c}}\}}} [|\mathbb{P}_F(C(\alpha)) - \mathbb{P}_G(C(\alpha))|]. \tag{5}$$

Such distance is symmetric, satisfying the triangle inequality as shown in appendix A.1. For the remainder of this section, we will show that the Generalized KS distance in eq. (5) meets the necessary $D_{\text{GKS}}(\mathbb{P}, \mathbb{P}) = 0$ and sufficient $D_{\text{GKS}}(\mathbb{P}_F, \mathbb{P}_G) > 0$ if $\mathbb{P}_F \neq \mathbb{P}_G$ conditions to consider it a metric. In the proof, we will rely on the probability density function of $\mathbb{P}$ with respect to a reference measure $\text{v}$, which we denote with $p : \mathcal{X} \to [0, \infty)$. Let

$$\Gamma_p(\lambda) := \{x : p(x) \geqslant \lambda\} \tag{6}$$

denote the *density level set of $p$ at level $\lambda \geqslant 0$* (also called the highest density region [21]), and let $\Pi_p := \{\Gamma_p(\lambda) : \lambda \geqslant 0\}$. The following observations about level sets will introduce the fundamental tools to prove the necessary and sufficient conditions for the generalized KS distance.

**Remark 1** (The silhouette [37]). *For any density $p$, the following holds*

$$p(x) = \int_0^\infty \mathbb{1}_{\Gamma_p(\lambda)}(x)\mathrm{d}\lambda, \tag{7}$$

*where $\mathbb{1}_C$ denotes the indicator function of a set $C$. The RHS of eq. (7) is called the* silhouette.

---

[2]In the general case, $C_{\mathbb{P},\mathcal{C}}(\alpha)$ at any given level $\alpha$ is not uniquely determined, i.e. there may exist several sets $C, C' \in \mathcal{C}$ s.t. $C \neq C'$ that satisfy the condition in eq. (3). For simplicity, we will call all such sets the (generalized) quantile sets at level $\alpha$ and write $C_{\mathbb{P},\mathcal{C}}(\alpha) = C$ and $C_{\mathbb{P},\mathcal{C}}(\alpha) = C'$ for all of them.

An immediate consequence of remark 1 is that $\Pi_p$ ordered with respect to $\lambda \geqslant 0$ fully characterizes $\mathbb{P}$, because $p$ does. Graphically, the silhouette is a multidimensional stack of level sets.

**Remark 2.** *Density level sets are minimum-volume sets [38]  The quantity $\mathbb{P}(C) - \lambda \mathrm{v}(C)$ is maximized over $\mathcal{A}$ by $\Gamma_p(\lambda)$, and thus if $\Gamma_p(\lambda) \in \mathcal{C}$, then $\Gamma_p(\lambda) = C_{\mathbb{P},\mathcal{C}}(\alpha)^3$ at level $\alpha = \mathbb{P}(\Gamma_p(\lambda)) = \int p(x)\mathbb{1}_{[\lambda,\infty)}(p(x))\mathrm{d}x.$*

Below, we present the fundamental theoretical result behind the proposed method, which restates Lemma 1.2. of Polonik [38].

**Theorem 1** (Necessary and sufficient conditions). *Let $\mathrm{v}$ be a measure on $(\mathcal{X}, \mathcal{A})$. Suppose that $\mathbb{P}_F$ and $\mathbb{P}_G$ are probability measures on $(\mathcal{X}, \mathcal{A})$ with densities (with reference measure $\mathrm{v}$) $f$ and $g$ respectively. Assuming that*

> *A.1 $\Pi_f \cup \Pi_g \subset \mathcal{C}$;*

> *A.2 $C_{\mathbb{P}_F,\mathcal{C}}(\alpha)$ and $C_{\mathbb{P}_G,\mathcal{C}}(\alpha)$ are uniquely determined[4] in $\mathcal{C}$ with respect to $\mathrm{v}$*

*the following two statements are equivalent:*

> *S.1 $\mathbb{P}_F = \mathbb{P}_G$;*

> *S.2 $D_{\mathrm{GKS}}(\mathbb{P}_F, \mathbb{P}_G) = 0.$*

See proof in Appendix A .

Meeting assumption **A.1** is a demanding challenge, almost equivalent to learning the target distribution. Below, we propose a relaxation of it, which we will use to show the validity of our method.

**Theorem 2** (Relaxation of assumption **A.1**). *Theorem 1 holds if assumption **A.1** is relaxed to the case that $\mathcal{C}$ contains sets that are uniquely determined with density level sets of $\mathbb{P}_F$ and $\mathbb{P}_G$ up to a set $C$ such that*

$$\forall_{C' \in 2^C} \ \mathbb{P}_F(C') = \mathbb{P}_G(C'), \tag{8}$$

*and let $r := \mathbb{P}_F(C) = \mathbb{P}_G(C)$, then the supremum in statement **S.2** is restricted to $[0, 1-r]$.*

See proof in Appendix A .

## 3  Kolmogorov–Smirnov GAN

For the remainder of the paper, we will consider $\mathbb{P}_F$ as the target distribution represented by a dataset $\{x_F\}$, and $\mathbb{P}_G$ as the approximate distribution that we want to train by minimizing the Generalized KS distance in eq. (5) with Stochastic Gradient Descent. We model $\mathbb{P}_G$ as a pushforward $g_{\theta\#}\mathbb{P}_Z$ of a simple (e.g., Gaussian, or Uniform) latent distribution $\mathbb{P}_Z$ supported on $\mathcal{Z}$, with a neural network $g_\theta : \mathcal{Z} \to \mathcal{X}$, parameterized with $\theta$, which we call the *generator*.

The major challenge in utilizing eq. (5) is the necessity of finding the $C_{\mathbb{P},\mathcal{C}}(\alpha)$ terms which is an optimization problem on its own. The idea that we propose in this work is to amortize the procedure by modeling the generalized quantile functions $C_{\mathbb{P}_F,\mathcal{C}}(\alpha)$ and $C_{\mathbb{P}_G,\mathcal{C}}(\alpha)$ with additional neural networks which have to be trained in parallel to the generator $g_\theta$. Therefore, our method is based on adversarial training [13], where optimization proceeds in alternating phases of minimization and maximization for different sets of parameters. Hence the name of the proposed method, the *Kolmogorov–Smirnov Generative Adversarial Network*.

### 3.1  Neural Quantile Function

The generalized quantile function defined in definition 1 is an infinite-dimensional vector function $C_{\mathbb{P},\mathcal{C}} : [0,1] \to C \in \mathcal{C}$. Such objects do not have an expressive, explicit representation that allows for gradient-based optimization. Therefore, we use an implicit representation inspired by density level sets in eq. (6). We propose to use *neural level sets* defined in definition 3 that are modeled by a neural network $c : \mathcal{X} \to \mathbb{R}$, which we will refer to as the *critic*.

---

[3]There may be other sets $C = C_{\mathbb{P},\mathcal{C}}(\alpha)$ but $\Gamma_p(\lambda)$ will certainly be one of them.

[4]In the sense defined in Polonik [38]

**Definition 3** (Neural level set). *Given a neural network $c : \mathcal{X} \to \mathbb{R}$, the neural level set at level $\lambda$ is defined as*[5]

$$\Gamma_c(\lambda) := \{x : c(x) \leqslant \lambda\}, \text{ and let } \Pi_c := \{\Gamma_c(\lambda) : \lambda \in \mathbb{R}\}. \tag{9}$$

Neural level sets are used, for example, in image segmentation [6, 20] and surface reconstruction from point clouds [3]. They fit our application because for computing the Generalized KS distance in eq. (5), the explicit materialization of generalized quantiles is not required as long as the probability measure can be efficiently evaluated on the implicitly specified sets. We set $\mathcal{C} = \Pi_c$, and thus $C_{\mathbb{P},\Pi_c}(\alpha) = \Gamma_c(\lambda_\alpha)$, with $\lambda_\alpha = \arg\min_{\lambda \in \mathbb{R}}\{\lambda : \mathbb{P}(\Gamma_c(\lambda)) \geqslant \alpha\}$. For a probability measure $\mathbb{P}'$ the following holds:

$$\mathbb{P}'\left(C_{\mathbb{P},\Pi_c}(\alpha)\right) = \mathbb{E}_{x \sim \mathbb{P}'}\left[\mathbb{1}_{(-\infty,\lambda_\alpha]}(c(x))\right], \tag{10}$$

which shows that the terms in eq. (5) under neural level sets can be Monte-Carlo estimated given samples from the respective distributions. Assumption **A.2** is satisfied by neural level sets by construction.

The formulation of the Generalized KS distance in eq. (5) includes two generalized quantile functions $C_{\mathbb{P}_F,\mathcal{C}}(\alpha)$ corresponding to target distribution $\mathbb{P}_F$ and $C_{\mathbb{P}_G,\mathcal{C}}(\alpha)$ corresponding to the approximate distribution $\mathbb{P}_G$. Both have to be modeled with the respective neural networks $c_{\phi_F}$ and $c_{\phi_G}$, where we use $\phi = \{\phi_F, \phi_G\}$ to denote the joint set of their parameters. In section 3.3, we show how to parameterize both critics with a single neural network. We set $\mathcal{C} = \Pi_{c_{\phi_F}} \cup \Pi_{c_{\phi_G}}$.

## 3.2 Optimizing generator's parameters $\theta$

The Generalized KS distance in eq. (5) is a supremum over a unit interval and two functions; thus, it can be upper-bounded as

$$D_{\mathrm{GKS}}\left(\mathbb{P}_F, \mathbb{P}_G\right) \leqslant \sum_{C \in \{C_{\mathbb{P}_G,\mathcal{C}}, C_{\mathbb{P}_F,\mathcal{C}}\}} \sup_{\alpha \in [0,1]} \left[|\mathbb{P}_F(C(\alpha)) - \mathbb{P}_G(C(\alpha))|\right]. \tag{11}$$

Next, we plug in $\mathcal{C} = \Pi_{c_{\phi_F}} \cup \Pi_{c_{\phi_G}}$ to eq. (11) and use eq. (10) to get generator's objective:

$$\mathcal{L}_g = \sum_{c_\phi \in \{c_{\phi_G}, c_{\phi_F}\}} \sup_{\lambda \in \mathbb{R}} \left[|\mathbb{E}_{x \sim \mathbb{P}_F}\left[\mathbb{1}_{(-\infty,\lambda]}(c_\phi(x))\right] - \mathbb{E}_{x \sim \mathbb{P}_G}\left[\mathbb{1}_{(-\infty,\lambda]}(c_\phi(x))\right]|\right]. \tag{12}$$

In practice, the expectations in eq. (12) are estimated on finite samples from the two distributions, i.e. $\{x_F\}$ mentioned before, and $\{x_G\}$ sampled from the approximate distribution $\mathbb{P}_G$ using the reparametrization trick to facilitate backpropagation of gradients. Therefore, the two terms become step functions in $\lambda$, and the supremum is located on one of the steps. That way, a line search on $\mathbb{R}$ reduces to a maximum over a finite set. To preserve the differentiability of the cost function calculated in this way, we apply Straight-through Estimator [4] in place of indication function $\mathbb{1}$. A schematic depiction of the process for a single critic is shown in fig. 1.

## 3.3 Optimizing critics' parameters $\phi$

By optimizing critics' parameters $\phi$, we want to satisfy assumption **A.1** so that Generalized KS distance becomes a metric. For the problem posed in such a way, we lack supervision, i.e., we do not know the target sets' shapes. However, we can reformulate the problem as an estimation of the density functions of the two considered measures $\mathbb{P}_F$ and $\mathbb{P}_G$ and use the obtained approximate density models to build level sets. We can constitute an optimization problem for such a task based solely on finite sets of samples, which we have for $\mathbb{P}_F$ and can arbitrarily generate from $\mathbb{P}_G$. As the estimator, we propose to use the Energy-based model (EBM) [43], which, thanks to the lack of constraints in the choice of architecture, can be very expressive while having favorable computational complexity at inference. To carry out EMB training effectively, we will introduce a new min-max game, the "min phase" of which will turn out to be the initial objective in eq. (5), and in this way, we will close the adversarial cycle.

Let the critic $c_{\phi_F}(x)$ serve as the energy function. The density given by the EBM is then $p_{c_{\phi_F}}(x) = \exp(-c_{\phi_F}(x))/Z_{c_{\phi_F}}$, where $Z_{c_{\phi_F}} = \int \exp(-c_{\phi_F}(x))\mathrm{d}x$ is the normalizing constant called partition

---

[5]Please note that the direction of the inequality in eq. (9) is opposite of the one in eq. (6) which is a convention that aligns the critic with the energy function of Energy-Based models.

**Algorithm 1:** Learning a generative model by minimizing Generalized KS distance.

---

**Input** : Target distribution $\mathbb{P}_F$; latent distribution $\mathbb{P}_Z$; generator network $g_\theta$; critic network $c_\phi$; number of critic updates $k_\phi$; number of generator updates $k_\theta$; score penalty weight $\beta$;

**Output** : Trained model $\mathbb{P}_G$ approximating $\mathbb{P}_F$;

**1 repeat**

**2**    **for** $i = 1$ *to* $k_\phi$ **do**

**3**      Draw batch $\{x\} \sim \mathbb{P}_F$ and $\{z\} \sim \mathbb{P}_Z$ ;         `// critic's inner loop`

**4**      $\mathcal{R}_c \leftarrow \frac{1}{|\{z\}|} \sum_{\{z\}} \|\nabla_x c_\phi(g_\theta(z))\|_2^2 + \frac{1}{|\{x\}|} \sum_{\{x\}} \|\nabla_x c_\phi(x)\|_2^2$;

**5**      $\mathcal{L}_c \leftarrow \frac{1}{|\{z\}|} \sum_{\{z\}} c_\phi(g_\theta(z)) - \frac{1}{|\{x\}|} \sum_{\{x\}} c_\phi(x)$;

**6**      Update $\phi$ by using $\frac{\partial(\mathcal{L}_c - \beta\mathcal{R}_c)}{\partial \phi}$ to maximize $\mathcal{L}_c - \beta\mathcal{R}_c$;

**7**    **for** $i = 1$ *to* $k_\theta$ **do**

**8**      Draw batch $\{x\} \sim \mathbb{P}_F$ and $\{z\} \sim \mathbb{P}_Z$ ;         `// generator's inner loop`

**9**      $\{c_F\} \leftarrow \{c_\phi(x) : \{x\}\}$ and $\{c_G\} \leftarrow \{c_\phi(g_\theta(z)) : \{z\}\}$;

**10**      $\{\lambda\} \leftarrow \{c_F\} \cup \{c_G\}$;

**11**      $\mathcal{L}_{g,F} \leftarrow \max_{\{\lambda\}} \left| \frac{1}{|\{z\}|} \sum_{\{c_G\}} \mathbb{1}_{(-\infty,\lambda]}(c_G) - \frac{1}{|\{x\}|} \sum_{\{c_F\}} \mathbb{1}_{(-\infty,\lambda]}(c_F) \right|$;

**12**      $\mathcal{L}_{g,G} \leftarrow \max_{\{\lambda\}} \left| \frac{1}{|\{x\}|} \sum_{\{c_F\}} \mathbb{1}_{(-\infty,-\lambda]}(-c_F) - \frac{1}{|\{z\}|} \sum_{\{c_G\}} \mathbb{1}_{(-\infty,-\lambda]}(-c_G) \right|$;

**13**      $\mathcal{L}_g \leftarrow \mathcal{L}_{g,F} + \mathcal{L}_{g,G}$;

**14**      Update $\theta$ by using $\frac{\partial \mathcal{L}_g}{\partial \theta}$ to minimize $\mathcal{L}_g$;

**15 until** *not converged*;

**16 return** $g_{\theta \#} \mathbb{P}_Z$

---

function. The standard technique for learning the model given target data distribution $\mathbb{P}_F$ is MLE, where the likelihood

$$\mathbb{E}_{x \sim \mathbb{P}_F}[\log p_{c_{\phi_F}}(x)] = \mathbb{E}_{x \sim \mathbb{P}_F}[-c_{\phi_F}(x)] - \log Z_{c_{\phi_F}} \tag{13}$$

is maximized wrt $\phi_F$. An unbiased estimate of the gradient of the second term can be obtained with samples from the EBM itself, typically achieved with MCMC sampling. Many approaches to avoid this expensive procedure have been described in the literature [43], and among them, the one based on adversarial training [23] is the most appealing to us. It introduces an auxiliary distribution $\mathbb{P}_{aux(F)}$, such that the gradient of eq. (13) wrt $\phi_F$ is approximated with the gradient of

$$\mathbb{E}_{x \sim \mathbb{P}_F}[-c_{\phi_F}(x)] - \mathbb{E}_{x \sim \mathbb{P}_{aux(F)}}[-c_{\phi_F}(x)]. \tag{14}$$

Consequently, an additional objective $\mathcal{L}_{aux(F)}$ must be introduced, the optimization of which will lead to the alignment of $\mathbb{P}_{aux(F)}$ and $\mathbb{P}_{c_{\phi_F}}$, where $\mathbb{P}_{c_{\phi_F}}$ denotes the probability distribution with density $p_{c_{\phi_F}}(x)$. We take an analogous approach to estimate $c_{\phi_G}(x)$.

When we (i) set $c_{\phi_G}(x) := -c_{\phi_F}(x)$, and (ii) repurpose $\mathbb{P}_G$ as $\mathbb{P}_{aux(F)}$ and $\mathbb{P}_F$ as $\mathbb{P}_{aux(G)}$, we show in appendix A.2 that the MLE objectives for the critics – now, denoted as $c_\phi$ – simplify as $\mathcal{L}_c = \mathbb{E}_{x \sim \mathbb{P}_G}[c_\phi(x)] - \mathbb{E}_{x \sim \mathbb{P}_F}[c_\phi(x)]$, which is then maximized in an adversarial game against the Generalized KS distance in eq. (5).

The standard approach for aligning the auxiliary distributions with their targets is to use the Kullback–Leibler divergence. We propose using the Generalized KS distance instead. We set $\mathcal{L}_{aux(F)} = D_{\mathrm{GKS}}(\mathbb{P}_G, \mathbb{P}_{c_\phi})$ and $\mathcal{L}_{aux(\mathbb{P}_G)} = D_{\mathrm{GKS}}(\mathbb{P}_F, \mathbb{P}_{-c_\phi})$. By analyzing these objectives in the fashion of section 3.2, we note that $\mathcal{L}_{aux(\mathbb{P}_G)}$ is the same as our original objective $D_{\mathrm{GKS}}(\mathbb{P}_F, \mathbb{P}_G)$ – which is symmetric – when we approximate sampling from $\mathbb{P}_{c_\phi}$ with the target distribution $\mathbb{P}_F$. Analogously for $\mathcal{L}_{aux(\mathbb{P}_G)}$ where sampling from $\mathbb{P}_{-c_\phi}$ is approximated with $\mathbb{P}_G$. Therefore, we have shown that the auxiliary objectives are already integrated into the adversarial game.

In practice, we find the *score penalty* regularizer of Kumar et al. [26], derived from the score matching objective, helpful to stabilize training. Therefore, we subtract it from $\mathcal{L}_c$ weighted by a hyperparameter $\beta$. In this way, we get a critic that is smoother and, therefore, generates regular level sets that facilitate optimization. We summarize the proposed training procedure in algorithm 1.

## 4 Discussion

In section 3.3, where we justify the choice of the critic's objective function, we refer to methods for training EBMs, which are approximate density distribution models. Thus, the reader can expect that our proposed critic $c_\phi$ in the limit of convergence of the algorithm will become a source of information about the density distribution of the target distribution $\mathbb{P}_F$ accompanying the model that generates samples $\mathbb{P}_G$. However, this does not happen as a consequence of the design choice (i), that is, the setup of $c_{\phi_F} = -c_{\phi_G} = c_\phi$. An EBM can only be equivalent to its inverse in the case of a uniform distribution. In addition, because of design choice (ii), during training, the critic is not evaluated outside of the support of $\mathbb{P}_F$ and $\mathbb{P}_G$ and, therefore, can reach arbitrary values there. Despite these observations, the Generalized KS distance present in our algorithm exposes sufficient conditions because of theorem 2.

The feature distinguishing KSGAN from other adversarial generative modeling approaches is that regardless of the outcome of the critic's inner problem, minimizing eq. (5) is justified because Generalized KS distance, despite not meeting assumption **A.1**, is a pseudo-metric [38]. For comparison, the dual representation of Wasserstein distance, used in WGAN [2] requires attaining the supremum in the inner problem.

The distances used for training generative models all fall into either the category of $f$-divergences $D_f(\mathbb{P}_F, \mathbb{P}_G) = \int_\mathcal{A} f\left(\mathrm{d}\mathbb{P}_F/\mathrm{d}\mathbb{P}_G\right)\mathrm{d}\mathbb{P}_G$ or integral probability metrics (IPMs) $D_\mathcal{F}(\mathbb{P}_F, \mathbb{P}_G) = \sup_{f \in \mathcal{F}} |\mathbb{E}_{x \sim \mathbb{P}_F} f(x) - \mathbb{E}_{x \sim \mathbb{P}_G} f(x)|$. The classical one-dimensional KS distance is an instance of IPM with $\mathcal{F} = \{\mathbb{1}_{(-\infty, t]} | t \in I\!R\}$ or $\mathcal{F} = \{\mathbb{1}_{G^{-1}(\alpha)} | \alpha \in [0, 1]\}$ when having access to the inverse CDF of one of the distributions based on eq. (2). One can see the Generalized KS distance from the perspective of IPM with $\mathcal{F} = \{\mathbb{1}_{C(\alpha)} | \alpha \in [0, 1] \,\&\, C \in \{C_{\mathbb{P}_F, \mathcal{C}}, C_{\mathbb{P}_G, \mathcal{C}}\}\}$. Assuming direct access to $C_{\mathbb{P}_F, \mathcal{C}}$ and $C_{\mathbb{P}_G, \mathcal{C}}$, for example when both $\mathbb{P}_F$ and $\mathbb{P}_G$ are Normalizing Flows [24, 34], measuring the distance comes down to a line search.

## 5 Related work

The need to generalize the KS test, and therefore distance, to multiple dimensions arose naturally from the side of practitioners who collected such data and wished to test related hypotheses. It was first addressed by Peacock [35], where a two-dimensional test for applications in astronomy was proposed. It involves considering all possible orders in this space and using the one that maximizes the distance between the distributions. A modification of this procedure has been proposed by Fasano and Franceschini [11] where only four candidate CDFs have to be considered, causing the test to be applicable in three dimensions, with eight candidates, under similar computational constraints. Chronologically, the following approach was the one on which we base our work, proposed in Polonik [38] but made possible by the author's earlier work [36, 37]. To the best of our knowledge, the first work that practically uses the theory developed by Polonik is Glazer et al. [12], which we recommend as an introduction to our work. It proposes applying the Generalized KS test based on the support vector machines for detecting distribution shifts in data streams.

As an instance of the adversarial generative modeling family, our work is related to all the countless GAN [13] follow-ups. We highlight those that study the learning process from the perspective of the distance being minimized. The work of Arjovsky and Bottou [1] provides a formal analysis of the heuristic tricks used for stabilizing the training of GANs. The $f$-GAN [33] proposes a unified training framework targeting $f$-divergences, which relies on a variational lower bound of the objective that results in the adversarial process. Approaches relying on the integral probability metric include FisherGAN [32], the Generative Moment Matching Networks [29] based on MMD, just like the later, more sophisticated MMD GAN [28], and finally the Wasserstein GAN (WGAN) [2] with the WGAN-GP follow-up [16] which shares common features with our work. Our maximum likelihood approach to fitting the critic results in the same functional form of the loss as WGAN(-GP) uses. In addition, the score penalty we use is similar to the gradient penalty of WGAN-GP.

## 6 Experiments

We evaluate the proposed method on eight synthetic 2D distributions (see appendix B.1 for details) and two image datasets, i.e. MNIST [27] and CIFAR-10 [25]. We compare against other adversarial

Table 1: Squared population MMD ($\downarrow$) between test data and samples from the methods trained on 65536 samples, averaged over five random initializations with the standard deviation calculated with Bessel's correction in the parentheses. The proposed KSGAN with $k_\phi = 1$ performs on par with the WGAN-GP trained with five times the budget $k_\phi = 5$. See appendix D.1 for qualitative comparison.

| Distribution | Method ($k_\phi$, $k_\theta$) | | |
| | GAN (5, 1) | WGAN-GP (5, 1) | KSGAN (1, 1) |
| --- | --- | --- | --- |
| swissroll | 0.00337 (0.001023) | 0.00029 (0.000119) | 0.00039 (0.000100) |
| circles | 0.00298 (0.001501) | 0.00027 (0.000215) | 0.00049 (0.000240) |
| rings | 0.00200 (0.001264) | 0.00013 (0.000082) | 0.00043 (0.000162) |
| moons | 0.00141 (0.000757) | 0.00035 (0.000136) | 0.00053 (0.000189) |
| 8gaussians | 0.00357 (0.002719) | 0.00035 (0.000248) | 0.00032 (0.000277) |
| pinwheel | 0.00166 (0.001451) | 0.00027 (0.000184) | 0.00040 (0.000086) |
| 2spirals | 0.00093 (0.000822) | 0.00027 (0.000191) | 0.00044 (0.000232) |
| checkerboard | 0.00143 (0.000899) | 0.00038 (0.000296) | 0.00086 (0.000468) |

methods, GAN and WGAN-GP, using the same neural network architectures and training hyper-parameters unless specified otherwise (see appendix C for details). All the quantitative results are presented based on five random initializations of the models. The source code for all the experiments is provided in anonymous code repository.

In all KSGAN experiments, we relax the maximum in line 11 and line 12 of algorithm 1 with sample average. In all experiments, we re-use the last batch of samples from the latent distribution (and target distribution in the case of KSGAN) from the critic's optimization inner loop as the first batch for the generator's optimization inner loop.

## 6.1 Synthetic distributions

Analyzing adversarial methods on synthetic, low-dimensional distributions is not popular. However, we conduct such an experiment because we are interested in whether the model generates samples from the support of the target distribution and how accurately it approximates the distribution. Working with small-dimensional distributions, we do not have to be as concerned about the curse of dimensionality when calculating sample-based distances, and we can visually compare the resulting histograms.

In table 1, we report the squared population MMD [15] between target and approximate distributions, computed with Gaussian kernel on 65536 samples from each distribution. Details about how we chose the kernel's bandwidth can be found in appendix B.1. GAN and WGAN-GP fail to converge with $k_\phi = k_\theta = 1$ (we do not report the results to economize on space); thus, we set $k_\theta = 5$ for them. The proposed KSGAN with $k_\theta = 1$ performs at a similar level to WGAN-GP, the better of the two former, despite using five times less training budget. We present additional results on the synthetic datasets in appendix D.1, which include performance with different training dataset sizes, non-default hyper-parameter setups for KSGAN, and histograms of the samples for qualitative comparison.

## 6.2 MNIST

We use the 50000 training instances to train the models, and based on visual inspection of the generated samples (reported in appendix D.2), we conclude that all the methods achieve comparable, high samples quality. To assess the quality of the distribution approximation, we use a pre-trained classifier on the same data as the generative models (details in appendix B.2). We run the same experiment on 3StackedMNIST [44], which has 1000 modes. We report the results in table 2.

In this experiment, we set the training budget for all methods to $k_\phi = 1$, $k_\theta = 1$ for a fair comparison. We find that all methods always recover all the modes with the standard MNIST target. However, GAN fails to distribute the probability mass uniformly between the digits. As the number of modes increases with the 3StackedMNIST target, GAN demonstrates its inferiority to other methods by losing 198 modes on average (four initialization cover approx. 985 modes, and one fails to converge, achieving only 98 modes). WGAN-GP and KSGAN consistently recover all the modes while being on par regarding KL divergence, which differs little between networks' initialization.

Table 2: The number of captured modes and Kullback-Leibler divergence between the distribution of sampled digits and target uniform distribution averaged over five random initializations with the standard deviation calculated with Bessel's correction in the parentheses. All the methods were trained with the same budget $k_\phi = 1$, $k_\theta = 1$. WGAN-GP and KSGAN cover all the modes in all experiments while demonstrating low KL divergence.

| Method $(k_\phi, k_\theta)$ | MNIST | | 3StackedMNIST | |
| --- | --- | --- | --- | --- |
| | # modes ↑ | KL ↓ | # modes ↑ | KL ↓ |
| GAN (1,1) | 10 (0.00) | 0.6007 (0.27550) | 808 (396.91) | 1.4160 (1.36819) |
| WGAN-GP (1,1) | 10 (0.00) | 0.0087 (0.00499) | 1000 (0.00) | 0.0336 (0.00461) |
| KSGAN (1,1) | 10 (0.00) | 0.0056 (0.00045) | 1000 (0.00) | 0.0362 (0.00534) |

Table 3: Inception Score (IS) and Fréchet inception distance (FID) metrics averaged over five random initializations with the standard deviation calculated with Bessel's correction in the parentheses. All the methods were trained with the same budget $k_\phi = 1$, $k_\theta = 1$. The scores for the training dataset are included in the top row, as "Real data" for reference. WGAN-GP and KSGAN perform similarly on average, while KSGAN exhibits lower variance between networks' initialization.

| Method $(k_\phi, k_\theta)$ | IS ↑ | FID ↓ |
| --- | --- | --- |
| Real data | 9.7256 | 5.8600 |
| GAN (1,1) | 2.1900 (0.08303) | 47.6419 (10.6864) |
| WGAN-GP (1,1) | 2.3464 (0.08397) | 43.0660 (6.73299) |
| KSGAN (1,1) | 2.3832 (0.04066) | 39.8881 (2.42623) |

### 6.3 CIFAR-10

We use the 50000 training instances to train the models and report the generated samples in appendix D.3. We train the models in a fully unconditional manner, i.e., not using the class information at all – contrary to many unconditional models that use class information in normalization layers. We quantify the quality of fitted models by computing the Inception Score (IS) [41] and Fréchet inception distance (FID) [18] from the test set and report the results in table 3 based on five random initializations. For reference, in the table, we include the IS of the training dataset and the FID between the training and test sets.

In this experiment, we set the training budget for all methods to $k_\phi = 1$, $k_\theta = 1$ for a fair comparison. All models fail to accurately approximate the target distribution, which is evident from a quantitative comparison in table 3 and a qualitative one in appendix D.3. KSGAN is characterized by the lowest variance between initializations among the methods considered.

## 7 Conclusions and future work

In this work, we investigated the use of Generalized Kolmogorov–Smirnov distance for training deep implicit statistical models, i.e., generative networks. We proposed an efficient way to compute the distance and termed the resulting model Kolmogorov–Smirnov Generative Adversarial Network because it uses adversarial learning. Based on the empirical evaluation of the proposed model, the results of which we report, we conclude that it can be considered as an alternative to existing models in its class. At the same time, we point out that many properties of KSGAN have not been studied, and we leave this as a future work direction.

Interesting aspects to explore are the characteristics of learning dynamics with the number of generator updates exceeding the number of critic updates, alternative ways to train the critic, and alternative representations of generalized quantile sets. The natural scaling of the Generalized KS distance may also prove beneficial regarding the interpretability of learning curves, learning rate scheduling, or early stopping. In addition, we hope that our work will draw the attention of the machine learning community to the Generalized KS distance, applications of which remain to be explored.

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

## A  Proofs

**Theorem 1** (Necessary and sufficient conditions). *Let* v *be a measure on* $(\mathcal{X}, \mathcal{A})$. *Suppose that* $\mathbb{P}_F$ *and* $\mathbb{P}_G$ *are probability measures on* $(\mathcal{X}, \mathcal{A})$ *with densities (with reference measure* v*)* $f$ *and* $g$ *respectively. Assuming that*

    *A.1* $\Pi_f \cup \Pi_g \subset \mathcal{C}$;

    *A.2* $C_{\mathbb{P}_F, \mathcal{C}}(\alpha)$ *and* $C_{\mathbb{P}_G, \mathcal{C}}(\alpha)$ *are uniquely determined[6] in* $\mathcal{C}$ *with respect to* v

*the following two statements are equivalent:*

    *S.1* $\mathbb{P}_F = \mathbb{P}_G$;

    *S.2* $D_{\mathrm{GKS}}\left(\mathbb{P}_F, \mathbb{P}_G\right) = 0$.

*Proof of Theorem 1.* The **S.1** $\implies$ **S.2** direction is trivial to show and works without satisfying the assumptions [38]. Therefore, we focus on showing that **S.2** $\implies$ **S.1**. Let

$$S_{\mathcal{C}}(\mathbb{P}) = \{(\mathrm{v}(C), \mathbb{P}(C)) : C \in \mathcal{C}\} \subset I\!\!R_+ \times [0,1], \tag{15}$$

and denote with $\Gamma(\lambda)$ the level set of density of $\mathbb{P}$ as defined in eq. (6), and let $\Pi := \{\Gamma(\lambda) : \lambda \geqslant 0\}$. Further, let $\tilde{S}_{\mathcal{C}}$ denote the least concave majorant [5] to $S_{\mathcal{C}}(\mathbb{P})$, that is, the smallest concave function from $I\!\!R_+$ to $[0,1]$ lying above $S_{\mathcal{C}}(\mathbb{P})$. $\tilde{S}_{\mathcal{C}}$ is supported on the generalized quantiles of $\mathbb{P}$ in $\mathcal{C}$, i.e. on the points $(\mathrm{v}(C_{\mathbb{P}, \mathcal{C}}(\alpha)), \mathbb{P}(C_{\mathbb{P}, \mathcal{C}}(\alpha)))$. Finally, let $\partial \tilde{S}_{\mathcal{C}}(\mathbb{P})$ be the intersection of the extremal points of the convex hull of $S_{\mathcal{C}}(\mathbb{P})$ with the graph of $\tilde{S}_{\mathcal{C}}$. Given $\Pi \subset \mathcal{C}$ which we assume in **A.1** for $\mathbb{P}_F$ and $\mathbb{P}_G$, and in the light of remark 2 we have that for any set $C$ such that $(\mathrm{v}(C), \mathbb{P}(C)) \in \partial \tilde{S}_{\mathcal{C}}(\mathbb{P})$ there is a level $\lambda$ for which $C = \Gamma(\lambda)$, and it is equal the left-hand derivative of $\tilde{S}_{\mathcal{C}}$ in the point $\mathrm{v}(C)$. From remark 1, we have that the silhouette fully characterizes $\mathbb{P}$, and therefore $\partial \tilde{S}_{\mathcal{C}}(\mathbb{P})$ does it as well.

Eventually, we conclude the proof with the observation that given **S.2**, under Lemma 2.1 of Polonik [38] (where **A.2** is utilized) we have that the extremal points of the convex hulls of $S_{\mathcal{C}}(\mathbb{P}_F)$ and $S_{\mathcal{C}}(\mathbb{P}_G)$ are the same points, thus $\partial \tilde{S}_{\mathbb{P}_F}(\mathbb{P}) = \partial \tilde{S}_{\mathbb{P}_G}(\mathbb{P})$, and finally $\mathbb{P}_F = \mathbb{P}_G$. $\qquad \square$

**Theorem 2** (Relaxation of assumption **A.1**). *Theorem 1 holds if assumption A.1 is relaxed to the case that* $\mathcal{C}$ *contains sets that are uniquely determined with density level sets of* $\mathbb{P}_F$ *and* $\mathbb{P}_G$ *up to a set* $C$ *such that*

$$\forall_{C' \in 2^C} \ \mathbb{P}_F(C') = \mathbb{P}_G(C'), \tag{8}$$

*and let* $r := \mathbb{P}_F(C) = \mathbb{P}_G(C)$, *then the supremum in statement S.2 is restricted to* $[0, 1-r]$.

*Proof of Theorem 2.* The statement in eq. (8) is equivalent to saying that $\mathbb{P}_F = \mathbb{P}_G$ on $(C, 2^C)$. Analogously to the proof of theorem 1 we can show that $\mathbb{P}_F = \mathbb{P}_G$ on $(\mathcal{X} \setminus C, 2^{\mathcal{X} \setminus C})$. By observing that probability measures are $\sigma$-additive, we conclude that $\mathbb{P}_F = \mathbb{P}_G$ on $(\mathcal{X}, \mathcal{A})$, and thus the result of theorem 1 holds. $\qquad \square$

---

[6]In the sense defined in Polonik [38]

## A.1 Generalized KS distance satisfies triangle inequality

Let us consider three probability measures $\mathbb{P}_F$, $\mathbb{P}_G$, and $\mathbb{P}_H$ on a measurable space $(\mathcal{X}, \mathcal{A})$.

$$
D_{\mathrm{GKS}}\left(\mathbb{P}_F, \mathbb{P}_H\right) + D_{\mathrm{GKS}}\left(\mathbb{P}_H, \mathbb{P}_G\right)
$$

$$
= \sup_{\substack{\alpha \in [0,1] \\ C \in \{C_{\mathbb{P}_F,c}, C_{\mathbb{P}_H,c}\}}} \left[|\mathbb{P}_F(C(\alpha)) - \mathbb{P}_H(C(\alpha))|\right] + \sup_{\substack{\alpha \in [0,1] \\ C \in \{C_{\mathbb{P}_H,c}, C_{\mathbb{P}_G,c}\}}} \left[|\mathbb{P}_H(C(\alpha)) - \mathbb{P}_G(C(\alpha))|\right]
$$

$$
\stackrel{(i)}{=} \sup_{\substack{\alpha \in [0,1] \\ C \in \{C_{\mathbb{P}_F,c}, C_{\mathbb{P}_H,c}, C_{\mathbb{P}_G,c}\}}} \left[|\mathbb{P}_F(C(\alpha)) - \mathbb{P}_H(C(\alpha))|\right] + \sup_{\substack{\alpha \in [0,1] \\ C \in \{C_{\mathbb{P}_H,c}, C_{\mathbb{P}_G,c}, C_{\mathbb{P}_F,c}\}}} \left[|\mathbb{P}_H(C(\alpha)) - \mathbb{P}_G(C(\alpha))|\right]
$$

$$
= \sup_{\substack{\alpha \in [0,1] \\ C \in \{C_{\mathbb{P}_F,c}, C_{\mathbb{P}_H,c}, C_{\mathbb{P}_G,c}\}}} \left[|\mathbb{P}_F(C(\alpha)) - \mathbb{P}_H(C(\alpha))|\right] + \left[|\mathbb{P}_H(C(\alpha)) - \mathbb{P}_G(C(\alpha))|\right]
$$

$$
\stackrel{(ii)}{\geqslant} \sup_{\substack{\alpha \in [0,1] \\ C \in \{C_{\mathbb{P}_F,c}, C_{\mathbb{P}_H,c}, C_{\mathbb{P}_G,c}\}}} \left[|\mathbb{P}_F(C(\alpha)) - \mathbb{P}_G(C(\alpha))|\right]
$$

$$
= \sup_{\substack{\alpha \in [0,1] \\ C \in \{C_{\mathbb{P}_G,c}, C_{\mathbb{P}_F,c}\}}} \left[|\mathbb{P}_F(C(\alpha)) - \mathbb{P}_G(C(\alpha))|\right] = D_{\mathrm{GKS}}\left(\mathbb{P}_F, \mathbb{P}_G\right)
$$

In (i), we use the fact that the supremum of absolute difference in distribution coverage is maximized with the generalized quantile function of one of them. In (ii), we apply triangle inequality for absolute value. Thus we have shown that $D_{\mathrm{GKS}}\left(\mathbb{P}_F, \mathbb{P}_H\right) + D_{\mathrm{GKS}}\left(\mathbb{P}_H, \mathbb{P}_G\right) \geqslant D_{\mathrm{GKS}}\left(\mathbb{P}_F, \mathbb{P}_G\right)$ which is the triangle inequality for the Generalized KS distance.

## A.2 Objective for the critic

Given two adversarial maximum likelihood objectives from Kim and Bengio [23], we (i) set $c_{\phi_G}(x) := -c_{\phi_F}(x)$, and (ii) repurpose $\mathbb{P}_G$ as $\mathbb{P}_{aux(F)}$ and $\mathbb{P}_F$ as $\mathbb{P}_{aux(G)}$, and show that:

$$
\frac{1}{2}\left(\mathbb{E}_{x \sim \mathbb{P}_F}[-c_{\phi_F}(x)] - \mathbb{E}_{x \sim \mathbb{P}_{aux(F)}}[-c_{\phi_F}(x)]\right) + \frac{1}{2}\left(\mathbb{E}_{x \sim \mathbb{P}_G}[-c_{\phi_G}(x)] - \mathbb{E}_{x \sim \mathbb{P}_{aux(G)}}[-c_{\phi_G}(x)]\right)
$$

$$
= \frac{1}{2}\left(\mathbb{E}_{x \sim \mathbb{P}_F}[-c_\phi(x)] - \mathbb{E}_{x \sim \mathbb{P}_G}[-c_\phi(x)] + \mathbb{E}_{x \sim \mathbb{P}_G}[c_\phi(x)] - \mathbb{E}_{x \sim \mathbb{P}_F}[c_\phi(x)]\right)
$$

$$
= \mathbb{E}_{x \sim \mathbb{P}_G}[c_\phi(x)] - \mathbb{E}_{x \sim \mathbb{P}_F}[c_\phi(x)].
$$

# B Experiments details

In this section, we provide additional details about experiments conducted in the paper that did not fit in the main text. All the models reported in the paper were trained under 12 hours on a single Nvidia GeForce GTX TITAN X GPU (12GB vRAM) with 32GB of RAM and 2 CPU cores. We report results based on 645 models trained, which amounts to 7740 GPU hours at most. We estimate that about three times as much computing time was used for preliminary experiments not reported in the paper.

## B.1 Synthetic

The synthetic 2D distributions are adopted from the official code of Grathwohl et al. [14] – `https://github.com/rtqichen/ffjord`. We randomly generate 65536 training and 65536 test instances from each distribution. In appendix D.1, we report the results of training the models with fewer instances but evaluated using the entire test set.

We choose the bandwidth of the Gaussian filter in squared population MMD as the median L2 distance between two samples, of 32768 instances each, from the simulator. The resulting values can be found in the code we provide with the paper.

Table 4: Architectures for synthetic 2D datasets.

| $z \in I\!\!R^8 \sim \mathcal{N}(0, I)$ |
| :---: |
| Linear(bias=True), $8 \to 512$ |
| ReLU |
| Linear(bias=True), $512 \to 512$ |
| ReLU |
| Linear(bias=True), $512 \to 512$ |
| ReLU |
| Linear(bias=True), $512 \to 2$ |

(a) Generator

| Linear(bias=True), $2 \to 512$ |
| :---: |
| LeakyReLU(slope=0.2) |
| Linear(bias=True), $512 \to 512$ |
| LeakyReLU(slope=0.2) |
| Linear(bias=True), $512 \to 512$ |
| LeakyReLU(slope=0.2) |
| Linear(bias=True), $512 \to 1$ |

(b) Critic

## B.2 MNIST

To detect the modes in the (3Stacked)MNIST experiments, we use a pre-trained classifier from PyTorch examples, trained for 14 epochs of the train set of the original MNIST dataset. We expect to find 10 and 1000 modes for the MNIST and 3StackedMNIST, respectively. We measure the KL divergence between the classifier's output and discrete uniform distribution for both distributions.

## B.3 CIFAR-10

We compute the Inception Score using the implementation from `https://github.com/sbarratt/inception-score-pytorch`. We compute the Fréchet inception distance using the implementation from `https://github.com/mseitzer/pytorch-fid`.

## C  Architectures and hyper-parameters

### C.1  Synthetic

For all of the methods and distributions, we use the same architecture, described in table 4, with spectral normalization [31] on linear layers for GAN. In all cases, we train the generator and critic with Adam($\beta_1 = 0.5$, $\beta_2 = 0.9$) optimizer with a constant learning rate of 0.0001, without L2 regularization or weight decay, for 128000 generator updates with batch size equal to 512. We use the standard loss for GAN, enforcing class 1 for real samples and 0 for generated samples. In WGAN-GP, we use 0.1 weight on gradient penalty (identified as a good value in preliminary experiments, which we do not report), and in KSGAN $\beta = 1.0$ as the weight for score penalty.

### C.2  MNIST

For the MNIST experiments, we use the DCGAN [39] architecture, without batch normalization layers, with 128-dimensional latent Gaussian distribution. For the 3StackedMNIST distribution, we increase the number of input and output channels for the critic and generator, respectively. We train the generator and critic with Adam($\beta_1 = 0.5$, $\beta_2 = 0.9$) optimizer with a constant learning rate of 0.0001, without L2 regularization or weight decay, for 200000 generator updates with batch size equal to 50. In the case of GAN for 3StackedMNIST, we use a learning rate of 0.001 (identified as a good value in preliminary experiments, which we do not report). We use the

flipped loss for GAN, enforcing class 0 for real samples and 1 for generated samples. In WGAN-GP, we use 10.0 weight on gradient penalty (identified as a good value in preliminary experiments, which we do not report), and in KSGAN $\beta = 1.0$ as the weight for score penalty.

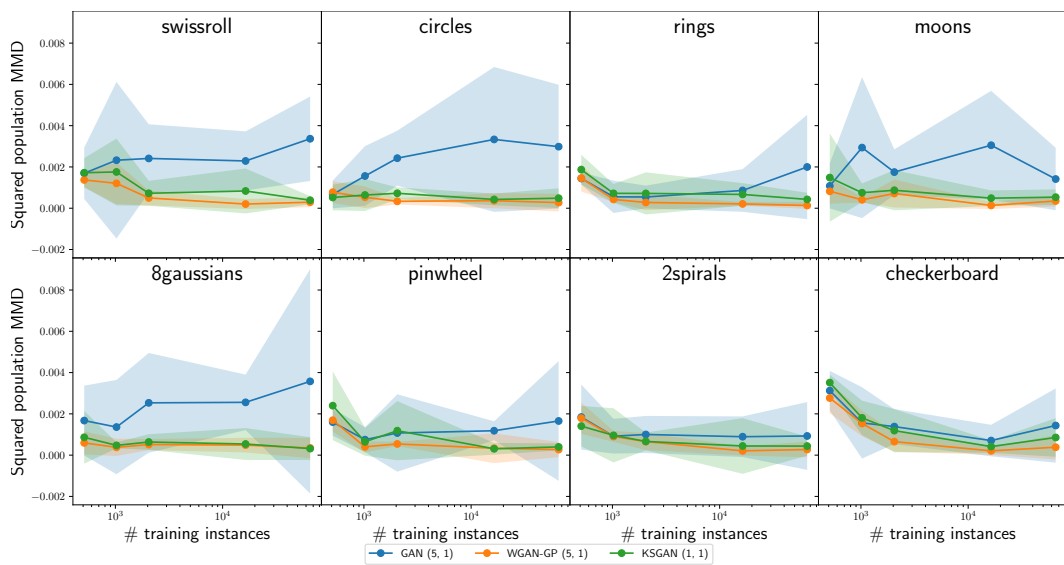

Figure 2: Squared population MMD between approximate and test distribution as a function of the number of training instances. Solid lines denote the average over five random initializations, and the shaded area represents the two-$\sigma$ interval. Best viewed in color.

## C.3  CIFAR-10

For the CIFAR-10 experiments, we use ResNet architecture from Gulrajani et al. [16]. We train the generator and critic with Adam($\beta_1 = 0.0$, $\beta_2 = 0.9$) optimizer with a constant learning rate of 0.0001, without L2 regularization or weight decay, for 199936 generator updates with batch size equal to 64. We use the

flipped loss for GAN, enforcing class 0 for real samples and 1 for generated samples. In WGAN-GP, we use 10.0 weight on gradient penalty (identified as a good value in preliminary experiments, which we do not report), and in KSGAN $\beta = 1.0$ as the weight for score penalty.

## D  Extended results

In this section, we report additional experiment results that did not fit in the main text. This includes materials allowing a qualitative comparison of the trained models.

### D.1  Synthetic data

In fig. 2, we report, extended relative to table 2 in the main text, a study of the quality of trained models as measured by the squared population MMD. Solid lines denote the average over five random initializations, and the shaded area represents the two-$\sigma$ interval. KSGAN performs on par with WGAN-GP while being trained with a five times less training budget. In fig. 3, we show the histograms of 65536 samples from the models (a single random initialization), with a histogram of test data in the first column for reference. For KSGAN, in addition to the configurations included in table 2, we include one with a training budget matching that of GAN and WGAN-GP, and one with a training budget reduced by two, where the critic is updated only every second update of the generator.

### D.2  MNIST

In fig. 4, we show samples from one of the random initializations reported in table 2 in the main text. All models demonstrate similar sample quality, while for GAN, the digit "1" is over-represented, which corresponds with the high KL in table 2.

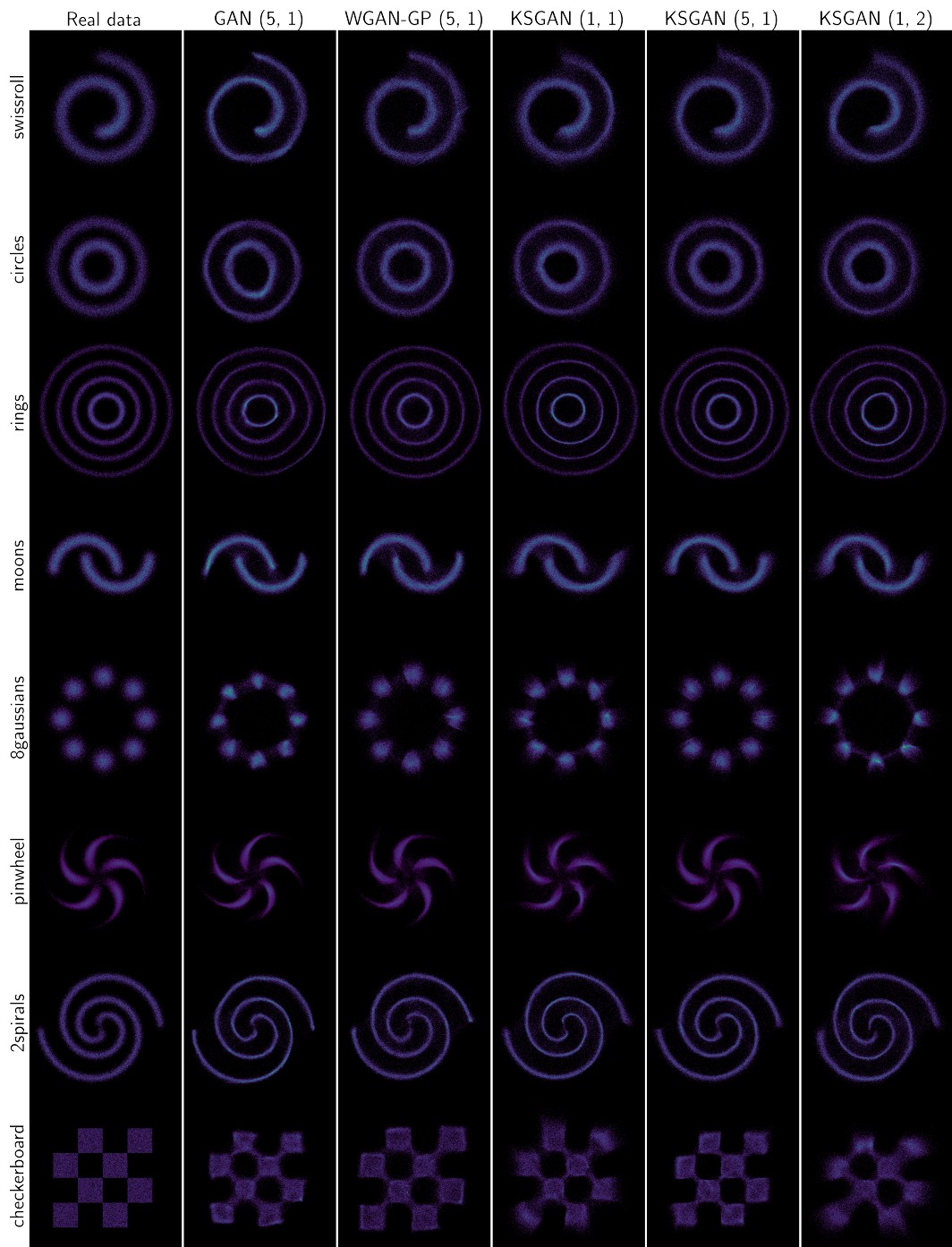

Figure 3: Histograms of samples from distributions denoted on the top. Heatmap colors are shared for all figures in each row. Best viewed in color.

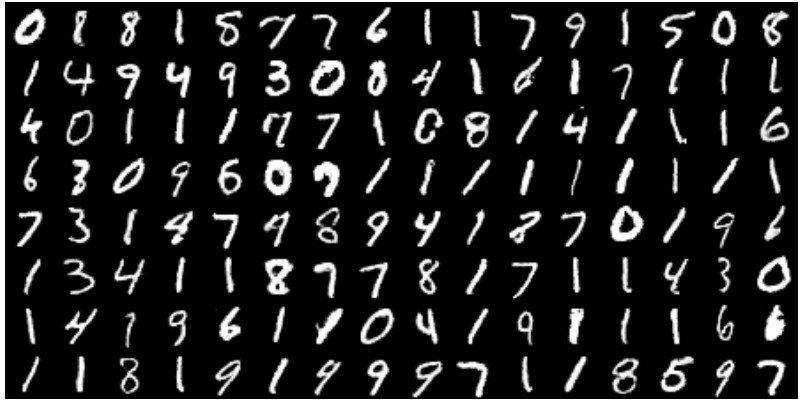

(a) GAN (1, 1)

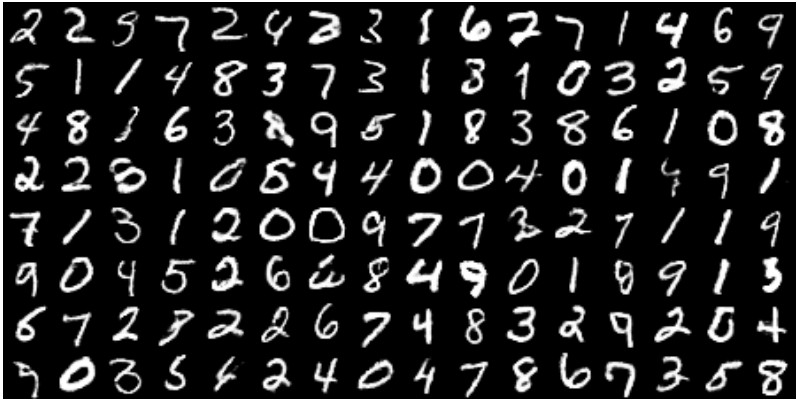

(b) WGAN-GP (1, 1)

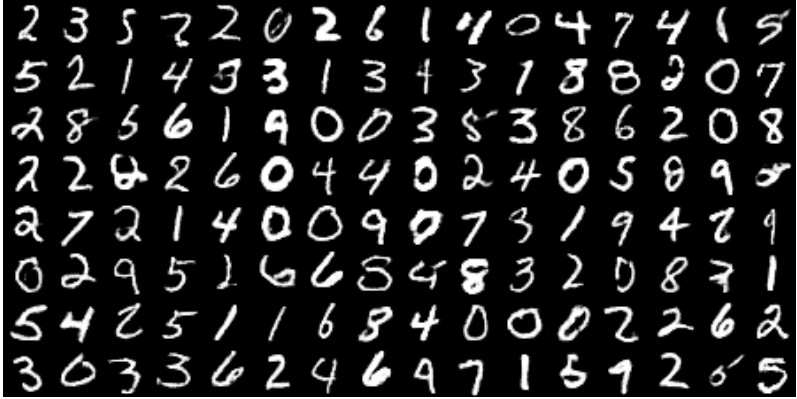

(c) KSGAN (1, 1)

Figure 4: Samples from the respective models trained on the MNIST dataset.

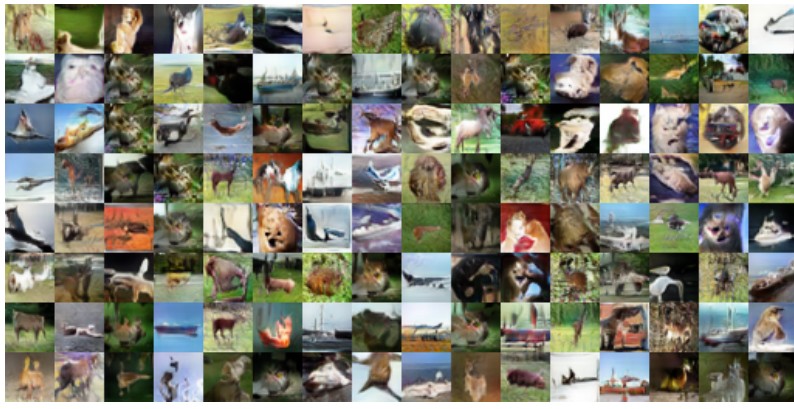

(a) GAN (1, 1)

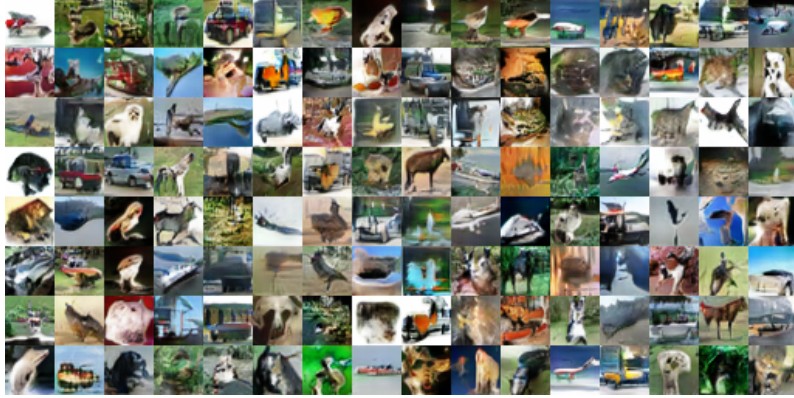

(b) WGAN-GP (1, 1)

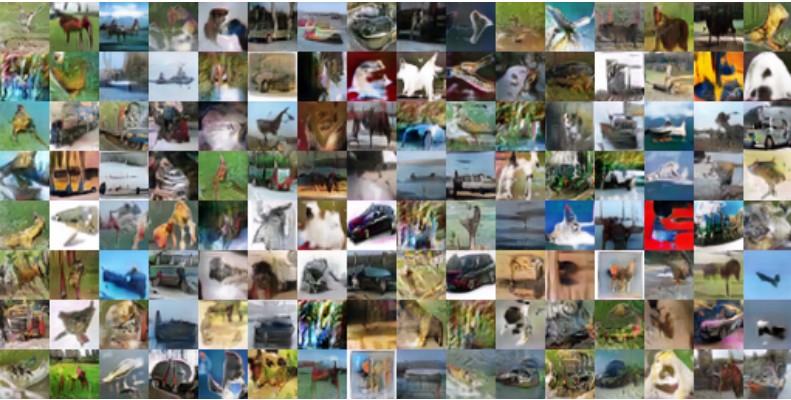

(c) KSGAN (1, 1)

Figure 5: Samples from the respective models trained on the CIFAR-10 dataset. Best viewed in color.

## D.3 CIFAR-10

In fig. 5, we show samples from one of the random initializations reported in table 3 in the main text. All models demonstrate similar, low sample quality.

