# OpenReview forum: "Kolmogorov–Smirnov GAN"
_NeurIPS.cc/2024/Conference — Submitted to NeurIPS 2024_

### Official Review · Reviewer_Yfup · 2024-07-10

**Soundness:** 3
**Presentation:** 3
**Contribution:** 2
**Rating:** 4
**Confidence:** 3

**Summary:**

This paper first generalizes the Kolmogorov–Smirnov (KS) distance from one-dimensional spaces to multidimensional spaces and proposes the Kolmogorov-Smirnov GAN, which formulates the generative model by minimizing the Kolmogorov-Smirnov (KS) distance. Theoretical results are also given in this paper and the experiments also show the superiority of stability during training, resistance to mode dropping and collapse, and tolerance to variations in hyperparameter settings.

**Strengths:**

I like the idea of generalizing the one-dimensional KS distance to multi-dimensional, which I had thought about before but failed to achieve. The motivation and writing are good, and the experiments of KSGAN seem to have achieved good results.

**Weaknesses:**

1. I'm skeptical about some parts of the theory. (See questions for details)
2. It seems that the advantages of KS distance over JS divergence and Wasserstein Distance are not explained.
3. The idea of reformulating a distance between distributions to a GAN model seems to be old and is now unlikely to attract readers' interest.
4. The experimental setup is relatively simple, only comparing with vanilla GAN and WGAN on Synthetic, MNIST, and CIFAR10 datasets
5. According to the experimental results, the advantages of KSGAN lie in the stability of training and resistance to mode dropping. A significant issue is that with current network architectures and training techniques, these two problems are rarely encountered.

**Questions:**

Q1. In Line 66-67, what's the meaning there are $2^d-1$ ways of defining a CDF on a d-dimensional space?

Q2. Definition 1 seems strange. What is the measure v? Is it a determined measure or any arbitrary measure? Is P the known CDF in Eq.3? What is its relationship with v?

Q3. Based on the definition of mapping, given function f: A->B, for each element a in A, there is always a unique element b in B corresponding to it. Then for $G^{-1}(\alpha)$ in Eq.2 and $C_{P,C}(\alpha)$, how is the uniqueness of the set guaranteed given $\alpha\in [0,1]$?

**Limitations:**

The authors have discussed the limitations.

---

> ### Author Rebuttal · Authors · 2024-08-05
>
> Thank you for your review. Please find answers to the posed questions below:
>
> Q1. “In Line 66-67, what's the meaning there are $2^d - 1$ ways of defining a CDF on a d-dimensional space?”
>
> Please find the answer to the question in reference [35] which we cite after the next sentence in L68 - middle of the third page of [35]. However, we will be happy to help you by guiding your intuition. For all the $d$ dimensions an order has to be chosen (thus $2^d$), but the ascending order in all dimensions is equivalent to the descending order in all dimensions (thus $-1$).
>
> Q2.
> - “What is the measure v? Is it a determined measure or any arbitrary measure?”
>
> In L82, just below Definition 1, we say that we consider $\mathrm{v}$ to be the Lebesgue measure. For additional discussion about $\mathrm{v}$ please see reference [10] which we cite in L75, just before the definition, or reference [38] of which Definition 1.1 is our Definition 1.
>
> - “Is P the known CDF in Eq.3?”
>
> $P$ is the probability measure.
>
> - “What is its relationship with v?”
>
> In [38] the author suggests considering $\mathrm{v}$ as the dominating measure for $P$.
>
>
> Q3. “[...] how is the uniqueness of the set guaranteed given $\alpha \in [0,1]$?”
>
> - Regarding $G^{-1}(\alpha)$: the assumption here is that $G(x)$ is continuous and strictly monotonically increasing. If the paper gets accepted, we will include this information
> - Regarding $C_{P,\mathcal{C}}(\alpha)$: we explicitly say that it is not unique in general (in footnote [2]), and intentionally use $\in$ operator in eq. (3). The uniqueness is necessary for Theorem 1 to hold (assumption 2). Our approach for parametrization of the generalized quantile functions - neural level set - satisfies this assumption. More generally, there is always such $\mathcal{C}$ that $C_{P,\mathcal{C}}(\alpha)$ is unique given $\alpha$.

---

> > ### Comment · Reviewer_Yfup · 2024-08-12
> >
> > Thanks for your rebuttal. I am still concerned about the weaknesses listed in my original comment and thus I maintain my score.

---

### Official Review · Reviewer_BXzn · 2024-07-12

**Soundness:** 2
**Presentation:** 3
**Contribution:** 1
**Rating:** 4
**Confidence:** 4

**Summary:**

This paper proposes a novel variant of the generative adversarial network that uses the Kolmogorov-Smirnov distance to align the generated distribution with the target distribution. This distance is calculated using the quantile function, which acts as the critic in the adversarial training process. Experiments are conducted on synthetic distributions and small image datasets to show that the proposed KSGAN performs on par with the existing adversarial methods.

**Strengths:**

1. The paper is well-presented and easy to follow.

2. The claims and methodology designs are well supported by theoretical analysis.

**Weaknesses:**

1. It is still unclear why we need another adversarial design based on KS distance. The vanilla GAN paper shows that the designed bi-level optimization process can already be seen as optimizing the distance between the generated and the target distribution. Then, what are the specific advantages KS distance can bring within the adversarial framework?

2. The experiments are merely conducted on synthetic datasets and small image datasets. It is unclear whether the proposed method can be adapted to larger-scale datasets or incorporated into more advanced frameworks like StyleGAN. Moreover, the compared baselines are limited to early works, and the experimental results of KSGAN are worse than those of WGAN-GP. Thus, I do not see many advantages of KSGAN in terms of the presented experiments.

**Questions:**

1. Could the authors explain or provide more evidence about line 40, "The Bayesian inference community has been reluctant to adopt adversarial methods"?

2. How can we guarantee the optimality of the learned quantile functions such that the estimated KS distance is reliable?

3. Again, with adversarial training, the original gan is already optimizing the distance between the two distributions. What are the benefits of optimizing in this way by KSGAN?

4. In Table 3, the performance of the models with IS around 2.4 and FID around 40 does not seem very successful in generative training. And the improvements seem minor.

**Limitations:**

Please see the discussions above.

---

> ### Author Rebuttal · Authors · 2024-08-05
>
> Thank you for your review. Please find answers to the posed questions below:
>
> Q1. “Could the authors explain or provide more evidence about line 40, "The Bayesian inference community has been reluctant to adopt adversarial methods"?”
>
> Our intention was to provide references [8] and [40] to support the claim, but if the paper gets accepted we will improve it. While [8] mentions GANs, none of the referenced methods uses GAN as the underlying generative model. In fact, [40] which was published in 2022 is to the best of our knowledge the first to study the use of GAN for SBI. The paper shows that GANs underperform compared to other established approaches. Moreover, despite the paper having multiple citations, the “GAN for SBI” direction is not being further explored.
>
>
> Q2. “How can we guarantee the optimality of the learned quantile functions such that the estimated KS distance is reliable?”
>
> In the finite resources (samples, optimization steps, non-infinitesimal learning rate, etc.) one cannot have such a guarantee, but this is the case for any learning algorithm, in particular GAN and WGAN. In the infinity limit, however, reference [23] supports the use of such adversarial training procedures as the one used by us, to minimize the KL divergence between the EBM and data distribution.
>
>
> Q3. “Again, with adversarial training, the original gan is already optimizing the distance between the two distributions. What are the benefits of optimizing in this way by KSGAN?”
>
> The original GAN paper [13] shows that under optimal discriminator, the generator’s objective is equivalent to the Jensen–Shannon divergence. However, in practice, training a vanilla GAN often fails, thus the many tricks described in [1] with their formal interpretation. Results in the literature, as well as our empirical evaluation of GAN, show that the model typically fails to accurately approximate the target distribution. Our empirical evaluation of the proposed KSGAN shows that the obtained approximations are more accurate, and this we consider as the benefit. In addition, as a potential future work direction, we see exploiting the theory of convergence rate of the classical KS distance in the generalized case. Any such results would be advantageous compared to the classical GAN based on JS divergence.
>
> Q4. “In Table 3, the performance of the models with IS around 2.4 and FID around 40 does not seem very successful in generative training. And the improvements seem minor.”
>
> The questions about results in Tab 3 are addressed in the global response.

---

> > ### Comment · Reviewer_BXzn · 2024-08-13
> >
> > Thank you for the response. Unfortunately, my concerns remain unaddressed. The authors claim that KSGAN can better approximate the target distribution based on experimental results. However, the results presented are basic and limited. Evaluating only on a basic dataset is quite restrictive, and I have not observed any significant performance improvement over state-of-the-art models. For real-world datasets, the authors only compared GAN and WGAN on CIFAR-10, but these methods are now somehow outdated. In the context of the current development of generative models, it is essential for papers in leading conferences to include SOTA algorithms and multiple large-scale datasets in their experiments. Regrettably, I cannot raise the score for this paper at this time.

---

> > > ### Author Response · Authors · 2024-08-13
> > >
> > > We want to thank the reviewer for this comment. We will work on providing results of using the proposed KS distance in advanced SOTA training pipelines like StyleGAN for future revisions of the papers.

---

### Official Review · Reviewer_L1Da · 2024-07-12

**Soundness:** 2
**Presentation:** 3
**Contribution:** 1
**Rating:** 3
**Confidence:** 4

**Summary:**

The authors introduce a generalized KS distance applicable to high dimensional spaces, formulate the corresponding dual problem, and use adversarial training to construct a generative model that minimizes the GKS between data and generated distributions.

The paper is well presented and appears technically correct through what I've seen, though I didn't check the proofs in details.

The main problem is that there's no clear motivation for why using the GKS is beneficial at all (either theoretically or practically). As such, despite being novel, I don't see any clear impact from the paper. Furthermore, the final algorithm is quite complicated and the results are fairly underwhelming, so at the end of the day the cons dramatically outweigh the pros of the newly introduced algorithm.

**Strengths:**

The paper is well written. It reads easily and it is clear what they want to do. The contribution of a generalized KS distance to multidimensional spaces and the algorithm to approximate it are to the best of my knowledge, novel.

**Weaknesses:**

The main problem I have with the paper is that I don't see any clear advantages of using the KS distance (gan) as a replacement of other distances like the Wasserstein one, or their GAN equivalent. The only mention of this, which should arguably be the most important thing in a paper introducing a new GAN, is in lines 224-228 of page 7. The authors claim there that they don't need to maximize the supremum in (5) which is false depending on how to interpret it, if you just take any set C in (5) you end up with |P_F(C) - P_G(C)| which is just measuring one moment for a given characteristic function, and far from being anything meaningful (and the same holding true for most IPMs). The results are also not particularly interesting to merit the claim that there's anything particularly different or benefitial on using this new formulation.

**Questions:**

The main suggestions I have is the following:
1) Take some time thinking why using the GKS is a better idea than using other distances between probability distributions.
2) Validate these claims with targeted experiments (for instance, on tractable distributions, without adversarial training).
3) Show that these properties translate to the adversarial setup, either theoretically or with experiments.

At the end of the day you're trying to convince readers that what you did is worth trying, so instead of focusing so much on mathematical details, consider more why a busy reader should care about this problem in the first place. What are the fundamental limitations of existing techniques you're trying to adress?

**Limitations:**

No clear motivation or benefit from using their algorithm.

---

> ### Author Rebuttal · Authors · 2024-08-05
>
> Thank you for your review. As there are no questions but there are suggestions instead, we would like to thank you for those. We will consider them in our future work.
>
> We would like to clarify one thing regarding the comment “The authors claim there that they don't need to maximize the supremum in (5) which is false[...]”. We never claim that attaining the supremum is not required. We claim (supported by the presented theoretical results) that the supremum is over a unit interval and two generalized quantile functions. In contrast, for Wasserstein distance, it is over all 1-Lipschitz functions. The challenge with KS distance is having access to the generalized quantile functions. The point of paragraph L224-L228 is to stress that no matter the quality of quantile functions, the computed “distance” is a pseudo-metric. But this is probably also the case for any IPM when the critic is not optimal. We will modify this section in the future revisions of the paper.

---

### Official Review · Reviewer_dLnS · 2024-07-15

**Soundness:** 3
**Presentation:** 3
**Contribution:** 2
**Rating:** 6
**Confidence:** 3

**Summary:**

The paper proposed a new kind of GAN training method called KS-GAN. The method is based on minimizing the Kolmogorov–Smirnov distance. The KSGAN updates the generator by minimizing an upper bound of the generalized KS distance. It updates the discriminator (or the critic network) by using energy-based model training with regularization terms. The KSGAN is a novel attempt to explore new approaches to train generative adversarial networks.

**Strengths:**

* (1) The paper studied using generalized KS distance to train GAN generators is a novel attempt to extend the GAN literature.

* (2) Some theoretical arguments about implementing the empirical KS distance using neural networks is novel yet constructive.

**Weaknesses:**

My main concern about the paper is its weak evaluation baselines and questionable practical usage:

* (1) Though the idea of using new objectives for training GAN generators is attractive, the practical usage of KSGAN seems questionable, especially for high-dimensional data. For instance, in the CIFAR10 generation experiment, the author compares KSGAN with WGAN-GP and Vanilla GAN, which have shown weak empirical performances. However, it is well-known that, for CIFAR10 data, the StyleGAN2-ADA[1] model is a strong baseline GAN model. I think it would strengthen the paper a lot if the authors could somehow show strong performances of KSGAN using StyleGAN2's architectures and implementation techniques. However, I do admit that such a requirement may be too tough for new methods.

* (2) The KSGAN's critic function is constructed with EBMs. However, even with regularization terms, energy-based models are well-known for poor scaling ability to high-dimensional data. This may prevent the practical usage of KSGAN for real-world high-dimensional data.

**Questions:**

Please see the weakness part.

**Limitations:**

The author has addressed the limitations.

---

> ### Author Rebuttal · Authors · 2024-08-05
>
> Thank you for your review. Please find answers to the posed questions below:
>
> W1. “[...] I think it would strengthen the paper a lot if the authors could somehow show strong performances of KSGAN using StyleGAN2's architectures and implementation techniques. [...]”
>
> We are aware that the best generative models relying on adversarial training use elaborate training procedures and complex architectures, consequently placing the optimization objective only as one of the elements of the method. Our intention was to consider only this aspect, in separation from the others. We leave employing the KS distance in training procedures such as StyleGAN as future work.
>
>
> W2. “[...] energy-based models are well-known for poor scaling ability to high-dimensional data.”
>
> Could you please provide a reference supporting the claim? To the best of our knowledge, EBMs are the most flexible and expressive framework. A non-exhaustive list of references:
> - Grathwohl, Will, et al. "Your classifier is secretly an energy based model and you should treat it like one." International Conference on Learning Representations. - every classifier is an EBM
> - Che, Tong, et al. "Your GAN is secretly an energy-based model and you should use discriminator driven latent sampling." Advances in Neural Information Processing Systems 33 (2020): 12275-12287. - in the same way, GAN’s discriminator is an EBM
> - Zhang, Dinghuai, et al. "Unifying generative models with GFlowNets and beyond." arXiv preprint arXiv:2209.02606 (2022). - one can even see GFlowNets (and Diffusion-based models) as EBMs

---

> > ### Comment · Reviewer_dLnS · 2024-08-12
> > **Thanks to the authors' rebuttal.**
> >
> > I thank the authors' efforts in rebuttal. I would like to raise my score to 6 (weak accept) to encourage authors on their innovation of using KS distance for training GAN generators.

---

### Author Rebuttal · Authors · 2024-08-05

We would like to thank all reviewers for their comments and suggestions. A recurring issue in all reviews is the insufficient performance of our method in the results we presented. Therefore, we will address it in a global response.

We would also like to emphasize that, according to the official "Call For Papers", contributions in the field of the theory are also within the thematic scope of this conference. All reviews confirm that our proposed method is innovative. No missing relevant literature has been reported, nor any shortcomings in the development of theoretical contributions – the few questions that have arisen are addressed in direct replies to the reviewers – and their practical instantiation. All the criticism comes down to the inconclusive experimental results. Below we show that in some experiments the results speak in favor of the proposed method. Deploying the advanced training techniques and architectures (like StyleGAN2) would only improve upon the reported performance, but it is beside the point since we want to focus on the bare-bone method (KS distance) and its evaluation.

All reviewers focused on CIFAR10 experiments, ignoring the other results. Meanwhile, in the case of synthetic distributions – which belong to the standard set of tests examining the accuracy of generative models [14] – the proposed KSGAN with a 5 times smaller computational budget, performs on par with the more expensive WGAN - not to mention GAN which is inferior to other methods in all cases. We want to emphasize that in L282 we mention that a WGAN with the same computational budget as KSGAN completely fails, while KSGAN with half of the budget is still able to provide sensible results (see Fig.3) . We would also like to point out that the KSGAN(5,1) with the same computational budget as WGAN-GP(5,1) outperforms all the other evaluated models (again, see Fig. 3). In the MNIST experiments, we show that KSGAN maintains a better balance between modes of distribution in the case of vanilla MNIST, and performs on par with WGAN 3StackedMNIST. These results highlight the practical relevance of the KSGAN, in addition to what we believe is a strong and novel theoretical contribution to the GAN literature.

In the case of CIFAR10 experiments, we found a bug in the implementation of calculating evaluation metrics. Furthermore, we identified discrepancies between our hyper-parameters and the reference implementation of WGAN-GP [16]. After making the changes, the results look as follows:


| Method | IS | FID |
|-----------|-----------|-----------|
| GAN(1,1) | 7.3523 (0.28921) | 34.8848 (2.85136) |
| WGAN(1,1) | 7.4690 (0.10070) | 28.5329 (0.93204) |
| KSGAN(1,1) | 7.4929 (0.08039) | 27.5118 (0.88830) |

Thus, KSGAN slightly outperforms WGAN, but the difference is not statistically significant.

---

> ### Comment · Reviewer_L1Da · 2024-08-12
> **Regarding comparisons and performance**
>
> One comment I want to make is that personally I don't think the main drawback is the lack of better experimental performance. In fact, I definitely welcome theoretical papers. However, I don't think theory is always useful and we as reviewers are also entrusted with the task of evaluating a theoretical contribution. To me a good theory paper usually solves or makes progress towards a problem that is of interest to the community (for instance, improving on a learning bound on a known scenario, or making a connection that could draw tools from other fields to solve something). I don't see that case happening here, there is little incentive (at least argumented in the paper) for using the KS distance instead of IPMs or other ones. Furthermore, the paper is from the abstract claiming that they want to introduce an algorithm to do something, so it makes sense for reviewers to judge said algorithm on the performance on doing said something. I don't really see here that the goal is to understand something better (nor how this paper would do such a thing), as theory papers are usually meant to do.

---

> > ### Author Response · Authors · 2024-08-13
> >
> > We want to thank the reviewer for this comment. We will try to establish learning bounds for GAN, WGAN-GP, and the proposed KSGAN for future paper revisions. As well as study other theoretical properties of the aforementioned methods.
> >
> > However, there are two things that we want to clarify:
> > 1. We do not suggest using KS instead of IPMs. We show that KS is an IPM.
> > 2. In terms of empirical results we do show improvements over the existing methods in terms of performance or by reducing computational demand.

---

> > > ### Comment · Reviewer_L1Da · 2024-08-13
> > >
> > > A few more details:
> > >
> > > 0) I did not suggest establishing learning bounds. Learning bounds can often be beside the point unless they are substantially different from one algorithm to another (or from one bound to another) in a way that's illuminating. If you do decide to go that route, try to think exactly where that difference may come from and why it's important.
> > > 1) My point of arguing why using KS instead of other IPMs is that if you introduce a new algorithm, in order for that to be a significant contribution, it's important to argue well why that algorithm has benefits that other algorithms do not.
> > > 2) I do understand that, though as other reviewers suggested, perhaps the evidence to that is not enough at this point.

---

### Decision · Program_Chairs · 2024-09-25

**Decision:**

Reject

**Comment:**

The paper proposes a deep generative model to train a GAN using the Kolmogorov-Smirnov (KS) distance as opposed to other distances like IPM.

After the rebuttal and discussion periods, 3 out of the 4 reviewers find that the paper is under the acceptance bar. The main criticism is that there is no clear incentive to use the KS distance in this case (its experimental performance isn't superior and there aren't clear advantages supporting this choice). Moreover, the experimental results are also somewhat limited (cf. Reviewer BXzn).

The authors argue that theoretical works are also welcome at NeurIPS. This is certainly the case, however it is not clear whether this paper should be judged as a theoretical contribution when its main contribution is to introduce an algorithm. Quoting Reviewer L1Da: "a good theory paper usually solves or makes progress towards a problem that is of interest to the community (for instance, improving on a learning bound on a known scenario, or making a connection that could draw tools from other fields to solve something). I don't see that case happening here [...] I don't really see here that the goal is to understand something better (nor how this paper would do such a thing), as theory papers are usually meant to do".